# What Is a Sustainable Coworking Space?

**Kolja Oswald ***  and **Xiaokang Zhao ***

Glorious Sun School of Business and Management, Donghua University, Shanghai 201620, China
*  Correspondence: 415039@mail.dhu.edu.cn or koloswald@gmail.com (K.O.); zxk@dhu.edu.cn (X.Z.)

**Abstract:** Coworking is a trend that is becoming increasingly popular and is often associated with sustainability. However, a lack of consensus exists on what a sustainable coworking space is. This study addresses this by investigating what is currently understood by a sustainable coworking space. Q-methodology is used to analyze 27 participants' subjective ideas about what a sustainable coworking space is, resulting in four distinct perspectives. The four perspectives are identified as follows: 1. "New Work", 2. "Resourceful Society", 3. "Incubator, and 4. "Environmental". These perspectives have distinct opinions on what important sustainability aspects in the context of coworking spaces are. Whilst some prioritize environmental and community factors, others have a mixed focus. Additionally, the four perspectives share some common beliefs. All of them believe in the importance of sustainable mobility, as well as in the moderate importance of encouraging their members to be socially responsible. These findings offer insight into the different understandings of coworking space sustainability. This is important because currently this field is under-researched, and a more systematic approach to sustainability in this field is needed. This research lays the foundation to do so and helps work toward a better understanding of coworking in a sustainability and innovative context.

**Keywords:** coworking space; sustainability; sharing economy; Q-methodology; open innovation

---

## 1. Introduction

Over the past decade, coworking has become increasingly popular. At present there are about 20,000 projected coworking spaces worldwide, and this number is expected to double by 2024 [1]. The rising trend of coworking shows a shift in the way humans work and is supported by the Fourth Industrial Revolution, that is changing how companies conduct business [2].

Coworking is known as being a practice within the sharing economy, as it encourages the sharing of office space, social space, and office infrastructure [3]. New technologies are creating more efficient consumption patterns whereby resources are shared [4]. As we move toward smart cities, the growth of coworking has the ability to shape industries and cities as a whole [5]. The transformational shift toward coworking may have the ability to increase economic welfare in a sustainable way [6,7]. Proponents of coworking often claim that sustainability is one of the core values of coworking, with a major global community of coworking spaces including it as a core value within its coworking manifesto [8,9]. It is believed that coworking spaces can be sustainable in their design and practice sustainable business models [10]. However, there is no consensus on what a sustainable coworking space entails.

It is reasonable to assume that a coworking space may not be inherently sustainable, and that there are certain factors that are especially relevant when creating a sustainable coworking space. Researchers have identified factors that may be important to a sustainable coworking space, such as environmental sustainability, sustainable mobility, and economic sustainability [11,12]. Yet, there is no clear answer or framework that constitutes what a sustainable coworking is. Thus, there is a

clear need to establish an understanding of what a sustainable coworking space is to establish a better understanding of the industry and its sustainability.

This paper contributes to this effort by elaborating on what a sustainable coworking space is believed to be. Using Q-methodology, this paper addresses the research question "What is a sustainable coworking space?" to develop four distinct perspectives on what a sustainable coworking space is. By uncovering these perspectives, the understanding of sustainability in the context of coworking spaces is expanded upon, offering a contribution to understanding the what and how of sustainable coworking spaces. The understanding of sustainability in the context of coworking is particularly important, as the Fourth Industrial Revolution is changing production and consumption. Whilst entrepreneurship has seemingly been in decline in the recent past, new phenomena such as coworking spaces may offer the ability to encourage an open innovation economy, potentially fostering new business [13,14]. As such, it is important to assess the sustainability of the coworking space phenomenon.

Q-methodology is suitable here because this research is exploratory in nature and seeks to uncover the subjective perceptions of the individuals that manage, own, and work with coworking spaces, and to understand their views. This investigation is therefore of an exploratory nature, and as such Q-methodology is applied. Q-methodology is a research methodology that is used to gather people's subjective perceptions on a certain topic [15]. This method was deemed applicable, as sustainability is a popular topic among coworking space owners and managers, whilst the current research on this topic is lagging behind. Unravelling these perspectives can help advance academia's understanding. As Q-methodology is a mixed qualitative and quantitative approach, giving the researcher a systematic framework to conduct research with, this methodology was seen as a suitable method with adequate academic rigor [16].

The structure of this paper is as follows. Section 2 covers the theoretical background on sustainability, coworking, and the sharing economy. Section 3 covers the methodology used within this study. Section 4 presents the results, and Section 5 presents the discussion. Lastly, Section 6 covers the conclusion, limitations, and suggestions for further research.

## 2. Theoretical Background

Sustainability research in the context of coworking spaces is still in its infancy. It has been acknowledged that, from an entrepreneurial and economic perspective, coworking spaces need to be sustainable [11]. It seems clear that internal factors within coworking spaces determine their economic sustainability [17]. From an economic perspective, the sustainability of a coworking space is partly determined by the level of continuous demand and occupancy of the space [18]. Furthermore, whilst business sustainability has been somewhat investigated, there is little existing research on other sustainability factors within coworking spaces. Environmental sustainability is an aspect that has been mentioned, yet no major findings have been reported [18]. Similarly, some research shows links between the climate of a coworking space and the sustainable business model outcomes of its member firms [12]. On the other hand, another study finds no causal relationships between the coworking space community and sustainability [19]. Another sustainability aspect that has been discussed in the context of coworking spaces is sustainable mobility. In theory, the creation of coworking spaces reorganizes the way people go to work. As this often occurs in urban environments, it may reduce the use of less sustainable means of transport and encourage the use of bicycles and public transport. Thus, Lejoux et al. identify sustainable mobility as a promising topic in this context. They also state that whilst there is a suggested link, there are no clear findings yet [15].

Thus, as of now there is a clear lack of understanding as to how coworking spaces can be sustainable. So far there are only fragmented findings, and there is no framework showing what a sustainable coworking space is. This is an important research gap to address because currently there is an inability to assess the sustainability of coworking spaces. Whilst coworking spaces have been discussed as generally having sustainable business models, no further analysis has been made in regards to different sustainability aspects of coworking space business models [20].

In order to identify what a sustainable coworking space business model is, it is important to define the term sustainable business model. Geissdoerfer et al. reviewed existing literature on sustainable business models and aggregated eight selected definitions, to define them as "business models that incorporate pro-active multi-stakeholder management, the creation of monetary and non-monetary value for a broad range of stakeholders, and hold a long-term perspective of stakeholders, and hold a long-term perspective." ([21], p. 403–404). Additionally, a sustainable business model may be viewed from a triple bottom line perspective, including an economic, an environmental, and a social layer [22]. This is important to consider, as coworking spaces may emphasize these layers differently, potentially resulting in various sustainable business models.

To further understand different types of sustainable business models, it is helpful to look at Bocken et al.'s eight archetypes of a sustainable business model framework [23]. This model gives insight into the theoretical and practical context of a sustainable business model. The eight archetypes are categorized into technological, social, and organizational groupings, and each archetype has multiple examples of sustainable business models [23]. Moreover, it gives a comprehensive overview of sustainable business models, suggesting that multiple archetypes may apply to coworking spaces. This framework was found to be particularly applicable to coworking spaces, as they may exhibit a range of different business models, potentially creating different sustainability implications [24]. Thus, Bocken et al.'s framework is considered particularly applicable in this context and is in part used as a theoretical foundation for this research [23].

Moreover, as coworking is a practice belonging to the sharing economy, sustainability aspects of the sharing economy may be relevant to coworking spaces' sustainability aspects [25]. The sharing economy consists of activities "where economic agents share economic objects together to create values" [13]. It is commonly accepted that business models within the sharing economy have the potential to create more sustainable consumption patterns, as they may have a positive economic, social, and environmental impact [26]. Whilst it is argued that sharing economy businesses may shift global production and consumption patterns toward a more sustainable future, Laukkanen and Tura found that, from a sustainability perspective, the sharing economy cannot be discussed as a whole, but a fragmented discussion is necessary [27,28]. They found that different business models within the sharing economy had distinct sustainability aspects, once again illustrating that there may be multiple types of sustainable business models within the context of coworking spaces.

Collaborative consumption, a major sharing economy phenomenon that is also observable in coworking spaces, is said to reduce unnecessary consumption, by reducing resource use, and creating a more sustainable system that addresses human needs [29]. It has been reported that collaborative consumption is in part motivated by the sustainability aspect itself, meaning that consumers choose this practice mainly because it seems more sustainable [30]. Similarly, technological openness and open innovation may boost technological innovations, which can increase the economic sustainability of coworking spaces [31]. This is because knowledge sharing is understood to be key in facilitating open innovation, which means that the sharing economy fosters open innovation as long as it facilitates knowledge exchange [32,33]. However, it is often unclear to what extent business models within the sharing economy are sustainable. At times, certain sectors of the sharing economy may have exhibited non-sustainable practices [34]. It is thereby clear that not all sharing economy practices are by default sustainable [35]. Whilst the direct economic transaction effects caused by the sharing economy are bound to be positive from a sustainability perspective, indirect economic effects and environmental effects are more complex and harder to assess [36]. It is therefore impossible to state whether a sharing economy phenomenon such as coworking is categorically sustainable or not, and a nuanced analysis is necessary. The fashion industry is an example of an industry where direct effects of collaborative consumption and resource sharing have been observable, yet where the sustainability effects of the sharing economy have been complex and hard to generalize [37]. Something similar may be observable in the coworking industry, where it may not be possible to assess sustainability on a meta level.

Curtis and Lehrer give insight into five semantic properties of the sharing economy that signal a sustainable approach. Firstly, the sharing economy is mediated by Information and Communications Technology. This means that technology offers the opportunity for the sharing economy to be scalable. Secondly, there is a motivation for non-ownership of the products. Thirdly, consumers only get temporary access to the products. Fourthly, the consumption of goods is rivalrous in nature. Lastly, the goods within the sharing economy are tangible [38]. These five semantic properties are useful in assessing the sustainability of parts of the sharing economy. Moreover, a large sustainability emphasis within the sharing economy is reported to be of environmental nature, with the majority of sustainability aspects being environmental [39]. Other reported sustainability dimensions within the sharing economy include the social, economic, spatial, and temporal dimensions [40]. Similarly, Daunoriene et al. identify the four major sharing economy sustainability perspectives as economic, environmental, societal, and technological [41].

Altogether, it is clear that sustainability is a commonly discussed theme within sharing economy research, yet there does not seem to be a homogenous approach [39]. Whilst coworking spaces commonly show key sustainable practices such as resource sharing, a consensus on homogenous sustainability aspects does not exist. This makes a sustainability analysis difficult. As the case-by-case assessment of sustainability is complex and nuanced, it is as of now impossible to objectively define a sustainable coworking space business model. Thus, exploratory research is needed to better understand the sustainability aspects of coworking spaces and to move toward an objective sustainability assessment of them.

This exploratory Q-methodology study uses Bocken et al.'s framework along with Q-methodology to develop a greater understanding of the interpretations of what sustainable coworking spaces are [24]. The analysis of subjective opinions of what a sustainable coworking space is will result in a greater understanding of the sustainability aspects emphasized within coworking spaces, helping us move closer to a transparent sustainability assessment of coworking spaces.

## 3. Methodology

In order to work toward a better understanding of what a sustainable coworking space is, this research paper uses Q-methodology to explore what coworking space owners, managers, and industry experts believe a sustainable coworking space is. Q-methodology is a mixed-method research technique that consists of both qualitative and quantitative research stages [42]. Q-methodology is applicable when studying subjectivity, as it helps understand people's subjective opinions, beliefs, and attitudes about a certain topic [17]. Q-methodology has proven to be a suitable research technique when it comes to sustainability research, as it offers a rigorous approach to subjectivity and can result in valuable policy implications [43].

Within the research field of sustainability, Q-methodology has been used several times. For example, one study investigated the subjective opinions of city officials about sustainability, identifying sustainability perspectives on urban design, economic development, and civic engagement [44]. Similarly, another paper used Q-methodology to investigate the perspectives of participants of a sustainable agriculture initiative, identifying technological and production function perspectives to be key sustainability perspectives [45]. In this case, Q-methodology is used to study the opinions that coworking space owners, managers, and experts have on what a sustainable coworking space is. Q-methodology studies follow a five-step approach that is adhered to in this research [16]. These five steps are:

1. Creating a concourse of communication known as the Q sample.
2. Selecting research participants known as P-set.
3. Performing the Q sort, where participants sort sampled statements.
4. Analyzing the Q sort using factor analysis.
5. Interpreting the factor analysis and identifying perspectives.

The detailed use of this five-step approach is discussed below.

*3.1. Q Sample*

The first step is to create a concourse. A concourse needs to enable participants to display different perspectives and, as such, contains all relevant topics and meanings associated with the topic [16]. The concourse may be collected from various primary and secondary research sources and should represent existing opinions and beliefs about the topic [46].

In this case, the concourse was created in the following systematic manner. First, Bocken et al.'s sustainable business model archetype framework was used as guidance [24]. Their sustainable business model archetype framework was used to sort sustainability statements into categories and sub-categories, with groupings used as categories and archetypes used as sub-categories.

This guaranteed a comprehensive concourse that covered a wide array of sustainability aspects of coworking space business models. Second, 42 written narratives from coworking spaces about sustainability were examined. These were retrieved from the websites of coworking spaces, as well as social media and blog posts made by coworking spaces. These texts were broken down, and statements were identified and sorted into the eight categories as statements. The requirement for each statement was that it appeared in more than two sources. Using written narratives to gather statements for the concourse is considered an appropriate technique, and using these narratives it was possible to gather over 60 sustainability statements [9]. Third, these statements were evaluated by reducing redundancies and asking six coworking space owners and managers for feedback and completeness. Finally, a sample of 35 sustainability statements was derived that was deemed comprehensive, because both secondary and primary sources did not offer any additional insights. Tables 1–3 shows the 35 sustainable statements sorted into Bocken et al.'s framework by category and sub-category [24].

**Table 1.** Sustainability statements 1–17.

| # | Category | Sub-Category | Statement |
|---|---|---|---|
| 1 | Technological | Material Productivity and Energy Efficiency | Installing an efficient lighting system |
| 2 | Technological | Material Productivity and Energy Efficiency | Installing an efficient water usage system |
| 3 | Technological | Material Productivity and Energy Efficiency | Minimizing paper waste |
| 4 | Technological | Material Productivity and Energy Efficiency | Minimizing plastic waste |
| 5 | Technological | Material Productivity and Energy Efficiency | Implementing an efficient HVAC system |
| 6 | Technological | Material Productivity and Energy Efficiency | Saving space and resources by creating an efficient office design |
| 7 | Technological | Material Productivity and Energy Efficiency | Complying with building sustainability codes (such as LEED) |
| 8 | Technological | Create value from waste | Implementing an effective recycling strategy |
| 9 | Technological | Create value from waste | Using reclaimed materials for construction of the coworking space |
| 10 | Technological | Create value from waste | Using reclaimed materials for furniture of the coworking space |
| 11 | Technological | Create value from waste | Using an existing building and renovating it, instead of building a new one. |
| 12 | Technological | Create value from waste | Implementing a composting strategy. |
| 13 | Technological | Substitute with renewables and natural processes | Using renewable energy (such as solar energy) to power the coworking space. |

**Table 1.** *Cont.*

| # | Category | Sub-Category | Statement |
|---|----------|--------------|-----------|
| 14 | Technological | Substitute with renewables and natural processes | Using Low Volatile Organic Compounds paint at the coworking space. |
| 15 | Technological | Substitute with renewables and natural processes | Creating a green interior coworking environment. |
| 16 | Technological | Substitute with renewables and natural processes | Creating a green exterior coworking environment. |
| 17 | Technological | Substitute with renewables and natural processes | Using eco-friendly cleaning products in the coworking space. |

**Table 2.** Sustainability statements 18–27.

| # | Category | Sub-Category | Statement |
|---|----------|--------------|-----------|
| 18 | Social | Deliver functionality rather than ownership | Flexible access to the coworking space around the clock. |
| 19 | Social | Deliver functionality rather than ownership | Sharing office infrastructure such as meeting rooms, printers, and internet access. |
| 20 | Social | Deliver functionality rather than ownership | Creating a communal kitchen and/or café for members to share. |
| 21 | Social | Deliver functionality rather than ownership | Creating an open office layout to increase communication among members. |
| 22 | Social | Adopt a stewardship role | Having the coworking space be near to public transport. |
| 23 | Social | Adopt a stewardship role | Offering healthy food choices to members. |
| 24 | Social | Adopt a stewardship role | Offering additional health-related services such as yoga classes or gym access |
| 25 | Social | Adopt a stewardship role | Sourcing organic beverages and/or snacks in the coworking space. |
| 26 | Social | Encourage sufficiency | Encouraging members to be socially responsible |
| 27 | Social | Encourage sufficiency | Encouraging members to walk, bike, or carpool to work. |

**Table 3.** Sustainability statements 28–35.

| # | Category | Sub-Category | Statement |
|---|----------|--------------|-----------|
| 28 | Organizational | Repurpose for society | Placing people over profit |
| 29 | Organizational | Repurpose for society | Creating a strong community within the coworking space |
| 30 | Organizational | Repurpose for society | Using the coworking space to support social causes |
| 31 | Organizational | Repurpose for society | Encouraging the formation of social enterprises |
| 32 | Organizational | Develop scale-up solutions | Encouraging collaboration among members of the coworking space. |
| 33 | Organizational | Develop scale-up solutions | Helping member firms/individuals expand their business |
| 34 | Organizational | Develop scale-up solutions | Focusing on long-term sustainability over shot-term profits. |
| 35 | Organizational | Develop scale-up solutions | Creating strong relationships with sustainable businesses outside of the coworking space. |

*3.2. P-Set*

Since Q-methodology is an intensive mode of analysis, purposive sampling should be used to maximize the study's quality in data [16]. Participants should be selected strategically, by finding participants that may have distinct opinions on the topic [47]. This is important because the sample size is comparatively small, and using an effective purposive sampling technique helps generate richer results.

In this study, the participants were selected strategically. Participants were selected according to two criteria. Firstly, they had to either have a top-management or ownership position within a coworking space, or they had to be industry experts within the coworking space industry. Secondly,

a specific emphasis was made on recruiting participants that had a focus on sustainability. This was done by reaching out to coworking spaces with specific sustainability mission statements. As a result, this sample can be considered as one that is skewed toward the belief in the importance of sustainability within the coworking context.

The final sample size of this study was 27. Critics of Q-methodology will argue that the often relatively small sample sizes are not sizeable enough for factor extraction. However, Arrindell and Van der Ende show in a study that even an observation to a variable of 1.3:1 can be effective, showing that smaller sample sizes can indeed be effective in factor analysis [48].

Of the 27 participants, 15 were coworking space managers, 10 were coworking space owners, and 2 were coworking space experts. The 2 coworking space experts were individuals that were owners of businesses that operated in the coworking industry for several years. All other 25 participants had more than a year's experience of managing and/or owning a coworking space.

All participants were informed beforehand that the participation in this study was completely voluntary and anonymous. A consent form was presented electronically to participants, which participants confirmed before commencing the Q-methodology study. All of the participants' data were kept anonymous at all times, and used solely for the purpose of this study.

### 3.3. Q-Sorting

The Q-Sort in this study was done using VQMethod [49]. VQMethod is an online platform developed to do Q-Sorting online. Prior to starting the Q-sort, participants were familiarized with the study and its objectives. Next, participants were asked to engage in a pre-sort. This pre-sort consisted of sorting the 35 statements into "Extremely important", "Not important at all", and "Neutral" categories, relating to the statements' importance in regard to coworking space sustainability. Whilst the pre-sort data was not recorded, this stage was included to help participants get an overview of all Q-Sort statements before proceeding with the main sort. This is a common step in Q-methodology to maximize the quality of the Q-Sort.

After the pre-sort, participants were then asked to partake in the main sort. As is visible in Figure 1, this Q-sort was designed in a quasi-normal grid that ranged from −4 to +4. Participants were asked to rank the 35 statements according to the prompt "Please sort the statements according to how important they are concerning sustainability of a coworking space. As you can see, the +4 grid denotes the most important factors of sustainability, whilst −4 denotes the least important. Please note that the vertical location of a statement has no meaning, therefore each statement within a value (+4, +3, . . . ) has the same allocated importance" on the −4 to +4 grid. As mentioned, the horizontal position was decisive, but not the vertical one. This means that all statements with the same grid value are assigned the same value irrespective of their vertical position.

Once participants completed the Q-sort, they were asked to reflect on the statements they ranked as most important (+4) and least important (−4). Lastly, participants were asked to answer a simple demographic survey.

### 3.4. Factor Analysis and Intepretation

After the participants completed the Q-sort, a factor analysis was done using PQMethod. PQMethod is a software created to analyze Q-methodology studies [50]. Using PQMethod, a principal component analysis in combination with varimax rotation were chosen as the factor analysis method. Principal component analysis and varimax are commonly used in Q-methodology research and are considered suitable factor analysis tools [16,51]. Using these tools, four factors were extracted, each resulting in a distinct perspective. As an additional analysis method, the Q-methodology researcher is encouraged to investigate the resulting perspectives for suitability, to ensure the quality of the data [52]. This was done, and all four perspectives were deemed relevant. Each factor was checked for clear distinguishing statements to ensure the quality of data extraction [53]. Upon factor extraction,

PQMethod was used to analyze the data using the various devices. For the interpretation, the results were once again compared to the framework created by Bocken et al. [24].

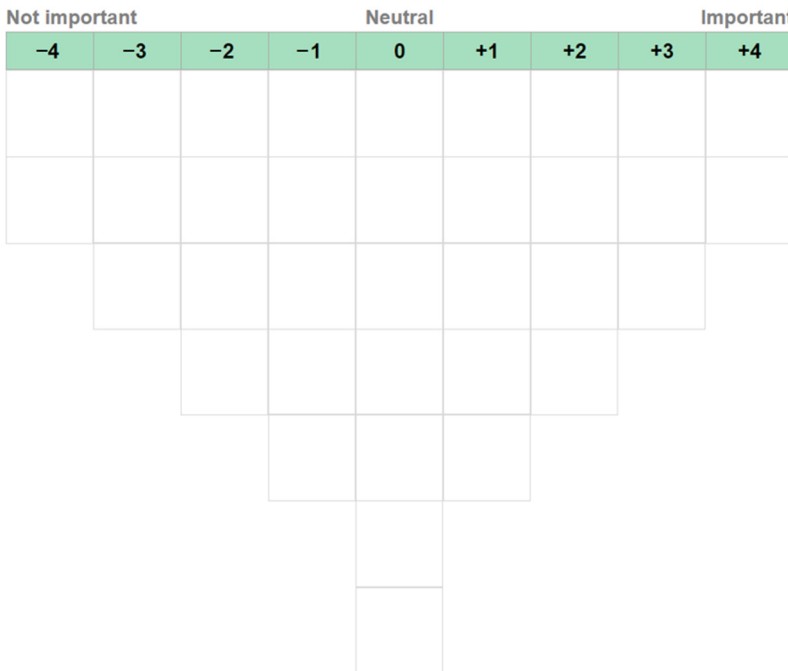

**Figure 1.** The Q−chart in VQMethod.

## 4. Results

The factor analysis resulted in four extracted factors, as seen in Table 4. The extracted factors as well as the key statistical indicators can be seen in Tables 4 and 5. The four factors had Eigenvalues of 7.1422, 4.7677, 2.5063, and 1.8434, which were all significantly above 1. The number of defining sorts per factor ranged from three to eight, with factor 4 having three defining sorts, which is higher than the requirement of two defining sorts per factor [54]. The total variance explained by the factors extracted equaled 60%, which shows that the variance of the principal components selected was considerably high, and higher than the minimum recommended amount of 50% [55]. Additionally, the composite reliability per factor ranged from 0.92 to 0.97, as can be seen in Table 5.

**Table 4.** Factor loadings including key statistical calculations.

| Participant # | Factor 1 | Factor 2 | Factor 3 | Factor 4 |
|---|---|---|---|---|
| 1 | 0.3055 | 0.4424 | −0.0796 | **0.643 *** |
| 2 | 0.2376 | **0.7503 *** | −0.0347 | 0.3453 |
| 3 | **0.6815 *** | 0.1000 | 0.1107 | 0.4450 |
| 4 | −0.2090 | **0.4676 *** | −0.0520 | 0.2557 |
| 5 | 0.2951 | −0.2118 | **0.6001 *** | 0.2232 |
| 6 | 0.4707 | −0.1459 | **0.6149 *** | 0.1517 |
| 7 | **0.6228 *** | 0.0597 | 0.5358 | 0.1466 |
| 8 | 0.2965 | −0.2035 | **0.7836 *** | 0.1888 |
| 9 | **0.6229 *** | −0.1783 | 0.1452 | 0.0523 |
| 10 | 0.1108 | 0.5468 | −0.0294 | **0.6733 *** |
| 11 | −0.3888 | **0.6030 *** | −0.3341 | 0.1920 |
| 12 | **0.5695 *** | 0.2070 | 0.2204 | −0.3442 |
| 13 | **0.7973 *** | −0.0765 | −0.0670 | 0.0054 |

**Table 4.** *Cont.*

| Participant # | Factor 1 | Factor 2 | Factor 3 | Factor 4 |
|---|---|---|---|---|
| 14 | 0.0865 | −0.1749 | **0.8436 *** | 0.2134 |
| 15 | −0.2955 | 0.4360 | 0.2297 | 0.5503 |
| 16 | 0.1330 | **0.7269 *** | −0.0471 | −0.1055 |
| 17 | **0.7346 *** | −0.0881 | 0.2018 | 0.0827 |
| 18 | 0.2941 | −0.2602 | **0.5976 *** | −0.1940 |
| 19 | 0.0897 | 0.1628 | **0.6100 *** | −0.0704 |
| 20 | 0.1158 | **−0.5430** | 0.0038 | 0.0216 |
| 21 | 0.2322 | −0.1576 | 0.2553 | 0.1750 |
| 22 | **0.7434 *** | 0.2750 | 0.2411 | −0.0382 |
| 23 | 0.1415 | −0.3056 | **0.5885 *** | −0.4356 |
| 24 | 0.1774 | −0.2102 | 0.1699 | **0.6863 *** |
| 25 | **0.7925 *** | 0.1241 | 0.1813 | 0.0957 |
| 26 | −0.1158 | 0.1791 | **0.8153 *** | −0.0921 |
| 27 | 0.2484 | **0.7968 *** | −0.1732 | −0.0373 |

* Indicates defining sort.

**Table 5.** Key statistical calculations regarding factor loadings.

| Statistical Calculations | Factor 1 | Factor 2 | Factor 3 | Factor 4 |
|---|---|---|---|---|
| % Explained Variance | 19 | 14 | 17 | 10 |
| Number of defining sorts | 8 | 6 | 8 | 3 |
| Average reliability coefficient | 0.8 | 0.8 | 0.8 | 0.8 |
| Composite reliability | 0.97 | 0.96 | 0.97 | 0.92 |
| S.E. of Factor Z-scores | 0.17 | 0.20 | 0.17 | 0.28 |
| Eigenvalues | 7.14 | 4.77 | 2.51 | 1.84 |

It is important to note that participants 15 and 21 did not load on any of the four factors effectively, as seen in Table 5. All other participants were successfully loaded on a single defining factor. The requirement for this was that the factor explained more than half the common variance, and that the loading was significant at $p < 0.05$. As is visible, participant 15 had a factor loading of 0.5503 on factor 4, but this did not explain more than half of the common variance, as participant 15's loading on factor 4 was smaller than the sum of the squares of all other loadings [56]. Thus, 25 of 27 sorts ended up in defining sorts, and the statistical criteria for a successful Q-methodology study were met. The four extracted factors were identified as the following perspectives:

1. Factor 1: "New Work" Perspective
2. Factor 2: "Resourceful Society" Perspective
3. Factor 3: "Incubator" Perspective
4. Factor 4: "Environmental" Perspective

*4.1. "New Work" Perspective*

The first factor was defined as the "New Work" perspective and was shared among eight participants. Figure 2 shows the four statements that participants of this perspective believed were most important, as well as the four statements they believed were least important when it comes to a sustainable coworking space.

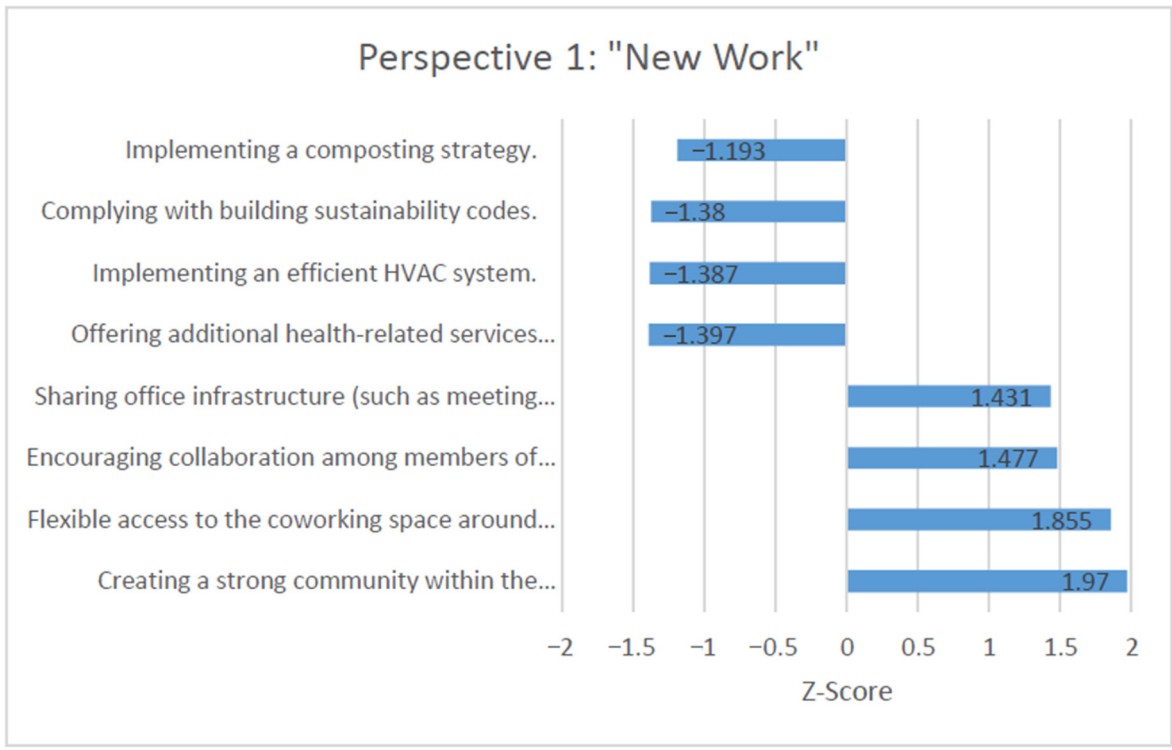

**Figure 2.** The four statements that the "New Work" perspective most strongly agrees with and disagrees with according to their Z-Scores.

The statements that participants agreed most with revolved around creating a strong community, having flexible access to the coworking space, encouraging collaboration, and sharing office infrastructure. As these traits somewhat describe the "co" in coworking, it seems that this sustainability perspective is about enforcing a new work paradigm that puts collaboration and community at the center of its business model. Participant 12's feedback on why creating a strong community is important was that "it is very important to support each other in order to build a sustainable entrepreneur ecosystem". This support can be encouraged by collaboration: as participant 13 points out, "connecting members to one another removes the owner/founder/staff as the main source of value". Thus, this perspective was coined as "New Work". Additionally, it is worth noting that of the five statements with the highest Z-Scores, three are of the "social" and two are of the "organizational" category of Bocken et al.'s framework [24]. This further illustrates the focus of this perspective.

As is visible in Figure 2, implementing a composting strategy, complying with building sustainability codes, implementing an efficient HVAC system, and offering health-related services were all deemed as particularly unimportant by participants within this group. This further underlines a narrow focus on the actual coworking process and a lack of focus on other aspects, especially environmental factors. In regards to composting, for example, respondent 9 stated "I don't understand why we need this", seeing no apparent use for it and respondent 12 further elaborated that they did not need one "because no café". These explanations show several things. Firstly, the importance of some sustainability statements clearly depends on certain factors within the coworking space such as the existence of a café. Secondly, the lack of understanding for why factors such as composting may not be relevant may be in part due to a lack of understanding of their utility in a sustainability context. Similarly, participant 25 stated that "I'm not sure there even are any sustainability codes". Thus, the lack of importance given to these factors may be in part caused by a lack of understanding. Yet, whilst some individuals within this group may have different reasons for ranking statements as unimportant, it is clear that what unifies this perspective is the strong belief in the importance of community and collaboration as sustainability aspects.

### 4.2. "Resourceful Society" Perspective

This perspective was defined as the "resource optimization" perspective and had six defining sorts. However, one of these six sorts was a negative loading. Participant 20 had a factor score of −0.5430. This suggests that participant 20 actually strongly rejected this perspective [57]. Therefore, it can be said that there were five defining sorts within this group that identified with this perspective. Figure 3 shows the statements that this group felt most strongly about. As is visible, the statements with the strongest positive Z-Scores evolved around optimizing resources from an environmental perspective.

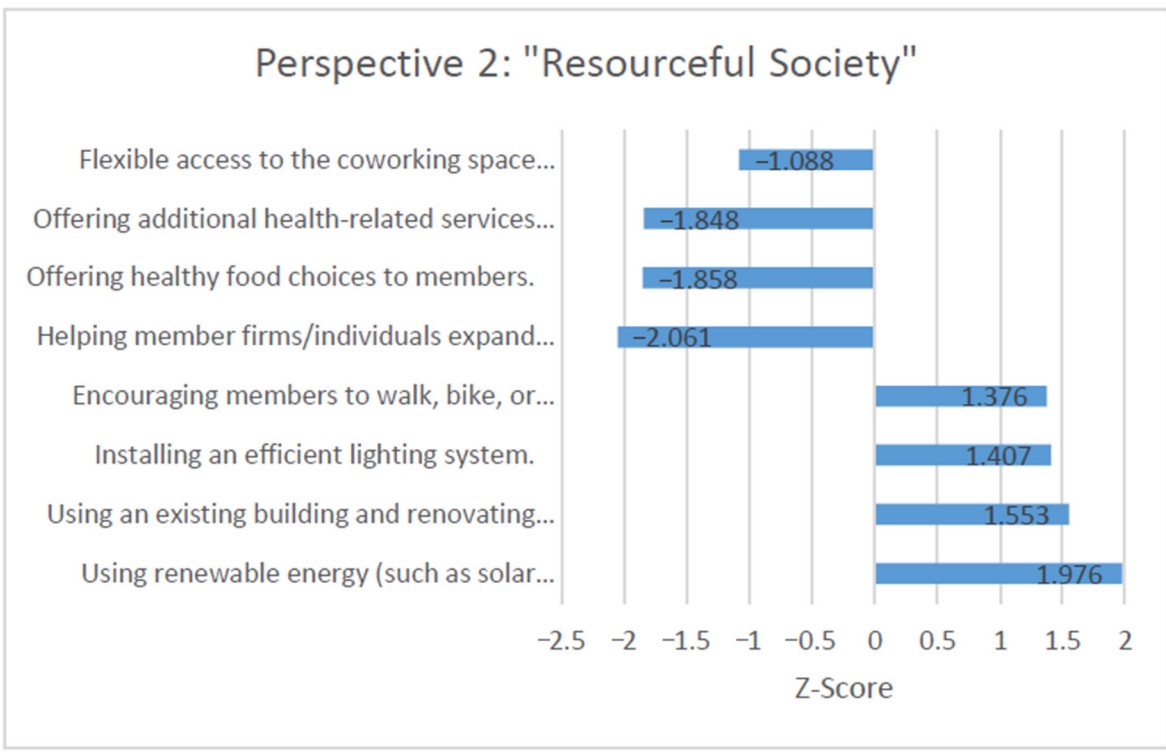

**Figure 3.** The four statements that the "Resourceful Society" perspective most strongly agrees with and disagrees with according to their Z-Scores.

The participants within this perspective found that using renewable energy was the single most important sustainability aspect. Participant 11 elaborated on why this statement is very important to him: "One of the most effective ways to reduce or eliminate the carbon emissions associated with operating a coworking space is to use 100% renewable energy, preferably on site with energy storage". The statement with the second highest Z-score within this perspective referred to using an existing building instead of building a new one. Participant 16 explained the importance of this: "Nowadays the expected lifespan of an office building is no more than 60 years and often a lot less. So, when reusing an existing building it has major positive impact on sustainability". The third and fourth highest Z-scores referred to installing an efficient lighting system and encouraging members to walk, bike, or carpool, which gives further insight into this perspective. It is clear that participants that are categorized into this perspective view a sustainable coworking space as one that optimizes its resource use in a sustainable manner. Of the five statements with the highest Z-score, three can be classified within Bocken et al.'s "technological" grouping, whilst two can be classified as "social factors" [24]. This further illustrates the emphasis on resource optimization with a social aspect.

This perspective seems to entail not placing a great importance on economic factors, as helping member firms expand their business and offering a flexible access to the coworking space ranked as some of the lowest statements in importance. In regards to helping member firms expand their business, participant 27 stated that it was "important for coworking but not for sustainability". This shows that,

at least for this participant, sustainability in terms of a coworking space is more resource-oriented than economic. Similarly, participant 4 stated that flexible access to a coworking space can actually be detrimental to sustainability, as "access to the coworking space around the clock increases the carbon footprint as for example light will have to be on". Moreover, the two health-related statements ranked very low in importance within this group. Participant 2 felt that offering healthy food choices to members simply "Isn't necessary." Participant 27 elaborated further on health-related services by stating "These services are not relevant to sustainability, they can be found elsewhere." Hence, it is clear that this perspective prioritizes direct resource optimization over economic and health-related sustainability aspects.

### 4.3. "Incubator" Perspective

Perspective 3 had eight defining sorts and was coined the "Incubator" perspective, due to its focus on helping the community and the businesses within it grow. Figure 4 shows this emphasis. Participants identified a strong community, a focus on long-term sustainability, encouraging collaboration, and placing people over profit as the statements with highest importance. Participant 19 outlined the importance of a strong community by stating that "This is the heartbeat of a coworking space and is essential for sustaining a viable space." Participant 26 explained that focusing on long-term sustainability helps strengthen this perspective by stating that a long-term mission would "change the mindset of the people, rather than saving the day". Participant 8 shines light on the importance of collaboration by stating that "What can be done in a group with someone who is just few meters from you who can offer you counseling, opinion and an idea. In other words. Work better together." Furthermore, the five statements with the highest Z-scores all belong into the "social" grouping of Bocken et al.'s framework [24]. Thus, the participants within this group believe in the importance of social sustainability. To them, a sustainable coworking space is one that focuses on a sustainable social setting, and less on environmental factors.

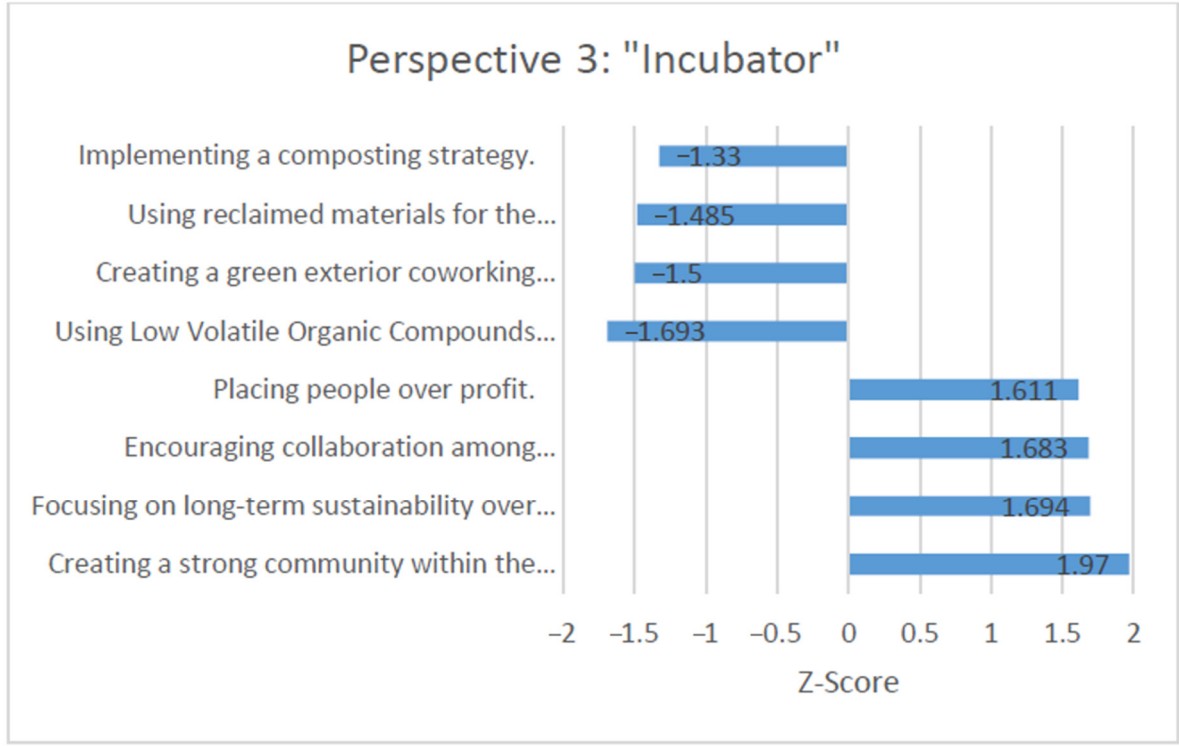

**Figure 4.** The four statements that the "Incubator" perspective most strongly agrees with and disagrees with according to their Z-Scores.

This is further corroborated by the four statements that have the lowest Z-score, which all relate to environmental aspects of sustainability. Participant 23 stated that using Low Volatile Organic Compounds paint results in "the least contribution to our mission". Hence, this participant did not necessarily state that this statement does not have a positive effect, but that it is not a significant one. Likewise, participant 14 stated that using reclaimed materials for the construction of the coworking space is not important due to "limitation and cost". It seems that in general, participants within this perspective do not dismiss the benefits of certain environmental sustainability aspects, but that they believe that the focus should be on creating a sustainable coworking space that helps grow a sustainable and supportive community.

*4.4. "Environmental" Perspective*

The fourth perspective had three defining sorts and was characterized as the "environmental" perspective. This is because all of the highest loading sustainability statements have a strong environmental focus, as seen in Figure 5. The highest-ranking statements were all heavily focused around creating a sustainable coworking space from an environmental perspective. Participant 1 emphasized that "Recycling is very important to reduce the carbon footprint of our coworking space", whilst participant 24 explained how composting is important for sustainability because "Food waste is a big contributor of greenhouse gases." Additionally, participant 24 stated that implementing an efficient lighting system is important because "lights use a lot of electricity and are constantly on during work hours. Therefore, an efficient system will save a lot of electricity." Furthermore, all of the top five Z-score statements fit into the "technological" grouping of Bocken et al.'s framework. This further illustrates the heavy focus on implementing sustainability aspects that have direct environmental effects and gives further insight into this perspective.

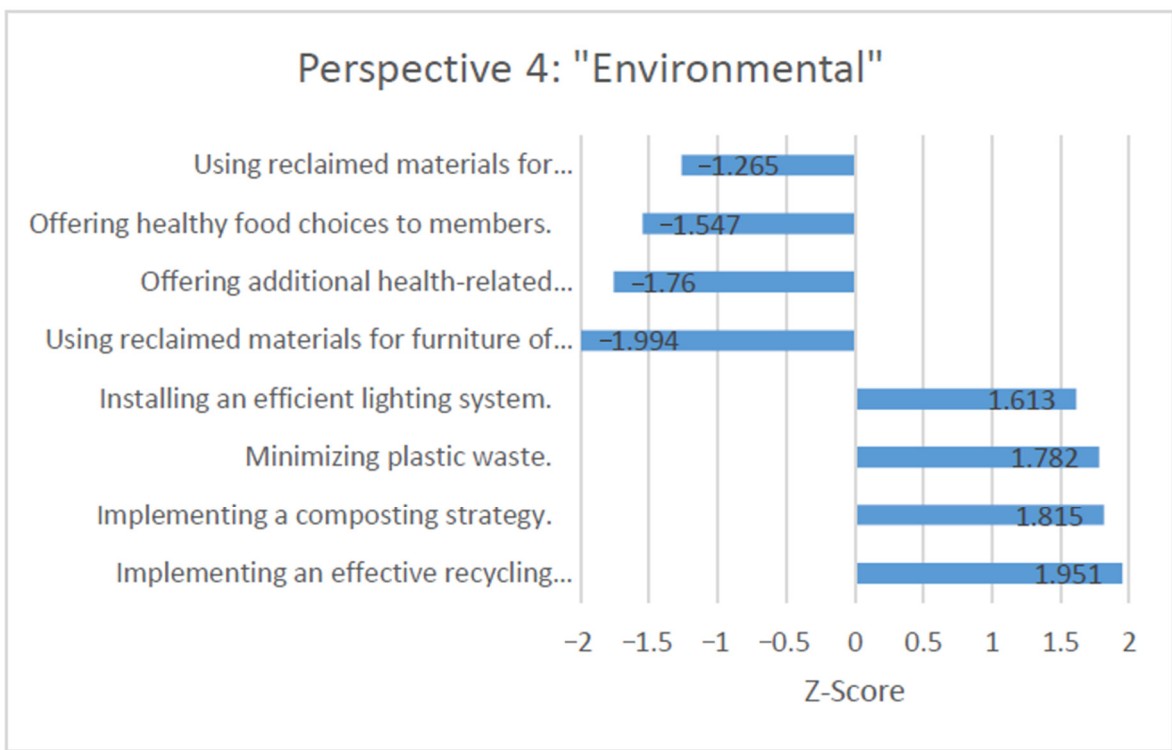

**Figure 5.** The four statements that the "Environmental" perspective most strongly agrees with and disagrees with according to their Z-Scores.

As is visible in Figure 5, participants within this perspective did not believe in the sustainability aspects of using reclaimed materials. Both statements that included this sentiment were sorted as

the two least important statements. This is peculiar, as both of these have an environmental impact, whilst many other statements within the concourse do not include an environmental perspective. Concerning this topic, participant 10 mentioned "I think you can use new materials if you need to", whilst participant 24 stated that using reclaimed materials for the space was "not an issue for me". This seems to suggest that there simply is not a big emphasis on the use of reclaimed materials, and that this perspective deems other environmental factors more essential. The other two of the four lowest-ranking statements were centered around health. Regarding healthy food choices, participant 1 stated that "I don't think the diet of our members makes us more or less sustainable." Furthermore, participant 1 stated that health-related services "are nice-to-have services, but again not important for sustainability of our space". Thus, this perspective heavily focuses on the environmental aspect of sustainability. The use of reclaimed materials is not part of this environmental focus, and the sustainability aspects of health are also not seen as priorities.

### 4.5. Similarities in Perspectives

The four perspectives have certain similarities. Table 6 shows the factor correlations between the four perspectives. The strongest communality exists between "New Work" and "Incubator", with a commonality of 0.3948. This is understandable, as the "New Work" perspective and the "Incubator" perspective both embody an emphasis on organizational sustainability aspects. Both perspectives include the sustainability archetypes "repurpose for society" and "develop scale-up solutions" from Bocken et al.'s framework in their top-five statements [24]. Furthermore, both factors share a Z-score of 1.97 for the statement "Creating a strong community within the coworking space".

**Table 6.** The factor correlation matrix.

| Factor | "New Work" | "Resourceful Society" | "Incubator" | "Environmental |
|---|---|---|---|---|
| "New Work" | 1.0000 | 0.1157 | 0.3948 | 0.3390 |
| "Resourceful Society" | 0.1157 | 1.0000 | −0.2307 | 0.3855 |
| "Incubator" | 0.3948 | −0.2307 | 1.0000 | −0.0957 |
| "Environmental" | 0.3390 | 0.3855 | 0.0957 | 1.000 |

The "Resourceful Society" and "Environmental" perspectives also have a significant commonality, amounting to 0.3855. As both the "Resourceful Society" and "Environmental" perspectives have a high focus on sustainability from an environmental perspective, this is understandable. In both perspectives, statements that identify with the "Create value from waste" and "Maximise material and energy efficiency" archetypes are within the top-five statements [24]. This shows the commonality between these two perspectives.

Regarding commonalities among all perspectives, Table 7 shows the top five consensus statements. What is particularly noticeable here is that all perspectives believe that a coworking space should be close to public transport in order to be sustainable. This suggests that sustainable mobility is a common sustainability aspect valued among coworking spaces and confirms previous research [15]. Furthermore, it seems that all perspectives find encouraging members to be socially responsible to be somewhat important. Lastly, it is clear that all perspectives believe that a green exterior environment is not important for coworking space sustainability.

**Table 7.** Top five consensus statements among the four perspectives.

| Statement | New Work | Resourceful Society | Incubator | Environmental |
|---|---|---|---|---|
| Encouraging members to be socially responsible | 1 | 0 | 1 | 0 |
| Creating a green exterior environment | −2 | −2 | −4 | −1 |
| Sourcing organic beverages and/or snacks | −3 | 0 | 0 | −2 |
| Creating a green interior environment | −1 | −2 | 0 | −2 |
| Having the coworking space be near to public transport | 3 | 3 | 2 | 1 |

*4.6. Differences in Perspectives*

Table 8 shows the top five disagreement statements among all perspectives. These statements give a good summary of the four distinct perspectives, as four of these five statements show top-ranking statements among a single perspective. The most disagreed upon statement is the statement with the second-highest Z-score in the "Environmental" perspective. The second most disagreed upon statement is the statement with the fifth-highest Z-score in the "Incubator" perspective. The third most disagreed upon statement is the statement with the second-highest Z-score in the "New Work" perspective. The fourth most disagreed statement is the statement with the second-highest Z-score in the "Resourceful Society" perspective. This shows a richness in disagreement, and suggests that the four perspectives are all distinct in their subjective opinion about the sustainability of a coworking space.

**Table 8.** Top five disagreement statements among the four perspectives.

| Statement | New Work | Resourceful Society | Incubator | Environmental |
|---|---|---|---|---|
| Implementing a composting strategy | −3 | 1 | −3 | 4 |
| Helping firms/individuals expand their business | 0 | −4 | 3 | −1 |
| Flexible access | 4 | −3 | −2 | −1 |
| Using an existing building and renovating it | 1 | 4 | −3 | −3 |
| Creating a strong community | 4 | −1 | 4 | 1 |

Table 9 further illustrates the uniqueness of each perspective by showing the top three distinguishing statements that each perspective strongly agrees with and strongly disagrees with. This is particularly useful in explaining the differences between the perspectives that have high commonalities.

Whilst "New Work" and "Incubator" share certain similarities, "New Work" is unique in its focus on creating a dynamic work environment, by offering flexible access, sharing resources, and creating an open office culture. This is different from "Incubator", where the focus is more on supporting its members and focusing on long-term sustainability. The two most distinguishing statements between these factors give further insight into the differences between these two perspectives. Whilst "New Work" prioritizes a flexible access to the coworking space, "Incubator" does not. Moreover, whilst "Incubator" prioritizes long-term sustainability, "New Work" does not.

**Table 9.** The top three distinguishing statements that each perspective most strongly agrees with and disagrees with.

| Perspective | Most Important | Least Important |
| --- | --- | --- |
| "New Work" | - Flexible access<br>- Sharing office infrastructure<br>- Open office layout | - Implementing HVAC<br>- Complying with building sustainability codes<br>- Using Low Volatile Organic Compounds |
| "Resourceful Society" | - Using renewable energy<br>- Using an existing building<br>- Encouraging members to bike, walk, carpool | - Helping member firms expand<br>- Creating a communal kitchen<br>- Creating an open office layout |
| "Incubator" | - Focusing on long-term sustainability<br>- Placing people over profit<br>- Helping firms expand their business | - Using Low Volatile Organic Compounds<br>- Creating a green exterior environment<br>- Minimizing paper waste |
| "Environmental" | - Implementing an effective recycling strategy<br>- Implementing a composting strategy<br>- Implementing HVAC | - Using reclaimed materials for furniture<br>- Saving space by creating an efficient office space<br>- Creating a strong community |

As previously mentioned, "Resourceful Society" and "Environmental" also have certain aspects in common. Table 9 shows that "Resourceful Society" is unique in its strong focus on renewable energy, whilst also focusing on using an existing building and encouraging members to use sustainable forms of transport. The "Environmental" perspective, on the other hand, is heavily focused on recycling, composting, and implementing an efficient HVAC system. The difference between the two is that "Resourceful Society" focuses more on social aspects such as long-term sustainability and helping member firms expand their businesses, whilst "Environmental" is heavily focused around environmental statements such as composting, minimizing plastic waste, and recycling.

Examining the differences between these four perspectives, it is helpful to look at Bocken et al.'s framework once more [18]. Figure 6 shows the four perspectives mapped out according to their emphasis on the three sustainable business model groupings: Technological, Social, and Organizational. This analysis tool shows how each perspective is distinct. The "Environmental" perspective occupies the technological area, the "Incubator" perspective is fully in the organizational area, whilst "New Work" and "Resourceful Society" operate in two areas at once. "New Work" has an organizational and social focus, whilst "Resourceful Society" has a technological and social focus.

THE FOUR COWORKING SPACE SUSTAINABILITY PERSPECTIVES

**Figure 6.** The four coworking space sustainability perspectives according to Bocken et al.'s three sustainability groupings [18].

## 5. Discussion

This study offers an insight into how coworking spaces view sustainability, particularly coworking spaces that put an emphasis on the topic of sustainability, as participants of this study felt strongly about the importance of sustainability in the coworking space context. The four perspectives identified in this study give an insight into four distinct perspectives on what a sustainable coworking space is. The "New Work" perspective embodies coworking spaces that view sustainability as a result of sharing resources as well as a thriving community. In a sense, this may be the perspective that is rooted in functionality, in the sense that this perspective prioritizes key sharing economy functions such as resource sharing and flexible access. The "Resourceful Society" perspective places a greater focus on environmental factors such as using sustainable energy, whilst still placing an importance on the social aspect of encouraging a sufficient use of resources as well as placing the coworking space close to public transport. The "Environmental" perspective, on the other hand, views sustainability from a strictly environmental perspective. This perspective seems to focus heavily on reducing the carbon footprint as a means to achieve a sustainable coworking space. On the contrary, the "Incubator" perspective views sustainability from an organizational perspective, whereby a coworking space is sustainable when it creates a long-term oriented community that helps each member grow.

As the research methodology applied investigates the subjective understanding of participants, it is important to state that none of these views are incorrect, they merely show a different emphasis. These findings suggest that each of these four perspectives offer insight into four different approaches to creating a sustainable coworking space. Simply because the "New Work" and "Incubator" perspectives show no strong emphasis on environmental factors within this research does not mean that they do not value the environment. Instead, their prioritized sustainability aspects may be ones that are crucial for them to thrive as a business, whilst environmental aspects may still be addressed.

What this research indicates is that it is important for coworking spaces to consider their business model and their identity in regard to sustainability. Whilst a coworking space may be able to address sustainability aspects holistically with a triple bottom line approach, addressing economic, social, as well as environmental issues, it seems clear that a specific business model requires certain sustainability aspects more than others. For instance, a coworking space that identifies with the "Incubator" perspective should focus more on fostering a community than a coworking space that identifies with the "Environmental" perspective.

Nonetheless, focusing on certain sustainability aspects does not simplify sustainability in general. Existing research shows that sustainability is often claimed, but not always implemented. This means

that whilst various coworking spaces may prioritize certain sustainability aspects over others, there is still a need to comply with standard sustainability aspects. The issue here is the lack of a unified framework when it comes to urban sustainability, and any effort may best be centered around the three sustainability pillars: environmental, social, and economic [58]. The 35 statements identified within this study, grouped into technological, social, and organizational categories, may also offer some value.

## 6. Conclusions

This research has answered the question "What is a sustainable coworking space?" in the eyes of coworking space owners, managers, and industry experts. In line with Q-methodology, the answer to this question is not objective, but subjective. It is found that there are four distinct perspectives on what constitutes a sustainable coworking space that each prioritize different sustainability aspects. The identification of these four perspectives is a major contribution to coworking space research because it shows that sustainability is defined differently within the coworking space industry. These different interpretations of sustainability suggest that there are multiple types of sustainable coworking space business models. By identifying these four perspectives, we move a step closer to being able to assess the sustainability of coworking space business models. This is important, as sustainability is often claimed in the industry, yet a lack of transparency exists as to what the meaning of sustainability is. This will help in understanding coworking spaces' functions within a sustainable economy and help us understand how coworking can be sustainable and contribute toward an open innovation system.

This research offers four approaches to a sustainable coworking space. There is the "New Work" perspective, that sees a sustainable coworking space as one that emphasizes organizational and social factors. People with this perspective believe that coworking space sustainability means flexible access to the space, as well as the sharing of resources. The "Resourceful Society" believes that a sustainable coworking space emphasizes sustainable and efficient energy use, whilst placing an importance on social factors. The "Environmental" perspective shares the emphasis on energy, but is more narrowly focused on environmental factors. For people with this perspective, reducing the carbon footprint seems to be the highest priority. Lastly, the "Incubator" perspective emphasizes the community within a coworking space. Here the view is that a collaborative community is absolutely key in developing a sustainable coworking space. Thus, this research identifies four distinct perspectives that are held within the coworking space industry. Whilst there are varying emphases on the sustainability statements, all perspectives agree that it is important for a coworking space to be closely located to public transport. Each perspective also believes that it is moderately important for the coworking space to encourage members to be socially responsible. However, as each perspective has its own opinion on the definition of sustainability, their common belief differs in interpretation. This once again confirms the contribution of this research.

Altogether, this research offers insight into the various ways a coworking space may be sustainable. The contribution of this research is key in order to understand the various meanings of sustainability in the context of coworking spaces. This helps make the discussion about coworking space sustainability less arbitrary and contributes to working toward a more holistic discussion of sustainability in this context.

*Limitations and Further Research*

The major limitation of this research is the small sample size (27). However, whilst this sample size is relatively small, it is still considered suitable for a Q-methodology study. From a statistical standpoint the study fulfilled the criteria, but it is also important to mention the researcher's subjective importance in a study like this. Within a Q-methodology study, the researcher's interpretation of the data is key, creating a clear limitation, as human judgment is somewhat biased. This applies to all Q-methodology studies, though, and is not a limitation of this particular study. Furthermore, as this study is of an exploratory nature, its role is to gain familiarity with the subjective opinions about coworking space sustainability. Lastly, it is important to consider that the sample selected had a bias

toward the importance of sustainability. A different sample may have yielded different results, yet the sample was chosen in this manner to extract different subjective perspectives on what a sustainable coworking space is. Altogether, the findings of this research are to be used with caution, and further research is needed to explore them.

The findings of this research offer various research opportunities. Firstly, it would be valuable to further research the four perspectives identified, to place them into context, and to move toward assessing them in terms of sustainability. As these perspectives are of a subjective nature, it will be important to move toward an objective sustainability assessment. Secondly, it would be useful to further research various business models among coworking spaces and to place these in a sustainability context. This could provide insights into sustainability aspects and be potentially expanded to the sharing economy in general. Lastly, there is a clear need for an integrated sustainability framework in the coworking space context. In the context of coworking spaces and the sharing economy, further research would be valuable. This could help assess sustainability, as well as offer tools on how to make businesses more sustainable.

**Author Contributions:** Conceptualization, K.O. and X.Z.; data curation, K.O.; formal analysis, K.O.; investigation, K.O.; methodology, K.O.; project administration, K.O. and X.Z.; supervision, X.Z.; visualization, K.O.; writing—original draft, K.O.; writing—review & editing, K.O. and X.Z. All authors have read and agreed to the published version of the manuscript.

**Funding:** This research received no external funding.

**Acknowledgments:** We wish to thank all the participants of this study for their efforts. We would also like to thank the editors and reviewers for their constructive feedback and help throughout.

**Conflicts of Interest:** The authors declare no conflict of interest.

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
