# Peer review of "What Is a Sustainable Coworking Space?"

_sustainability, doi:10.3390/su122410547_

Round 1
Reviewer 1 Report
- Please provide more evidences of the literature review argument. The literature review is necessary for you to clarify the “contribution” of your study
- Coworking spaces is sustainability. I think the author has to provide evidences “Another sustainability aspect that has been discussed in the context of coworking spaces is sustainable mobility. Whilst, Lejoux et al. identify sustainable mobility as a promising topicin this context, they unravel a great need to further frame the definitions of coworking spaces, sustainability, as well as mobility. Thus, while there is a suggested link, there are no clear findings”. Perhaps, this is related to sharing ?
- Table 4 should only show those validate result
- What are the contribution ? The major defect of this study is the debate or Argument is not clear stated in the introduction session. Hence, the contribution is weak in this manuscript.
- Sample size 27 can work on EFA and Q ?
- Please make sure your conclusions' section underscore the scientific value added of your paper, and/or the applicability of your findings/results, as indicated previously
Author Response
Dear Reviewer,
We thank you for taking the time to give us feedback to improve our paper. We have spent the last few days improving the paper to the best of our ability and are grateful for your constructive feedback. Below is the point-by-point response. We submitted one manuscript with highlighted changes and one without.
- Thank you for pointing this out. We have added several segments within the literature review to reflect your feedback. They are as follows:
- Line 77 to 85 we added the following section:
“In theory, the creation of coworking spaces reorganizes the way people go to work. As this occurs often occurs in urban environments, it may reduce the use of less sustainable means of transport and encourage the use of bicycles and public transport. Thus, Lejoux et al. identify sustainable mobility as a promising topic in this context, they also state that whilst there is a suggested link as of now there are no clear findings [8]. Thus, as of now there is a clear lack of understanding as to how coworking spaces can be sustainable. So far there are only fragmented findings and there is no framework showing what a sustainable coworking space is. This is an important research gap to address because currently there is an inability to assess the sustainability of coworking spaces.”
- Line 105 to 106 we added the following section:
“Thus, Bocken et al’s framework is considered particularly applicable in this context and is in part used as a theoretical foundation for this research”[18].
- Line 153 to 156 we added the following section:
“This exploratory Q-methodology study uses Bocken et al’s framework along with Q-methodology to develop a greater understanding of the interpretations of what sustainable coworking spaces are [18]. The analysis of subjective opinions of what a sustainable coworking space is will result in a greater understanding of the sustainability aspects emphasized within coworking spaces, helping us move closer to a transparent sustainability assessment of coworking spaces.” - Yes, thank you for helping us clarify that point. The new section reads as follows (line 77 to 81),
“In theory, the creation of coworking spaces reorganizes the way people go to work. As this occurs often occurs in urban environments, it may reduce the use of less sustainable means of transport and encourage the use of bicycles and public transport. Thus, Lejoux et al. identify sustainable mobility as a promising topic in this context, they also state that whilst there is a suggested link as of now there are no clear findings [8].” - Thank you for this feedback. We split Table 4 into two. Now Table 4 only shows the factor loadings, whilst Table 5 shows the other key statistical calculations. We also thought about removing non-defining sorts, but after reviewing other published Q-methodology studies we found that it is the norm to show all the respondents and their loadings, if they did not represent a defining sort.
- Thank you for pinpointing this weakness. We added following section to address your concerns (line 44 to 50):
“This paper thereby contributes by elaborating on what a sustainable coworking space is believed to be, by asking respondents what they believe a sustainable coworking space is. Using Q-methodology this paper addresses the research question “What is a sustainable coworking space?”, to develop four distinct perspectives on what a sustainable coworking space is. By uncovering these perspectives, the understanding of sustainability in the context of coworking spaces is expanded upon, offering a contribution to understanding the what and how of sustainable coworking spaces. Q-methodology is suitable here because this research is exploratory”. - Thank you for your question. Small sample sizes in Q-methodology are considered the norm. Most samples in Q-methodology studies tend to have a size between 10-30 (Chang et al, 2019). There are several Q-methodology studies published in the Sustainability journal that have similar sample sizes or smaller. In regards to sample size and factor identification we mention in our research, “However, Arrindell & Van der Ende show in a study that even an observation to variable of 1.3:1 can be effective, showing that smaller sample sizes can indeed be effective in factor analysis [41].”. Thus, the sample size of 27 was sufficient to identify 4 factors. Also, the statistical requirements for each factor were met as each had an Eigenvalue of above 1.0 and over 2 defining sorts.
- Thank you for helping us improve this section by showing us that we didn’t underscore the value of our paper enough. We added the following sections:
- Line 548 to 556:
“The identification of these four perspectives is a major contribution to coworking space research because it shows that sustainability is defined differently within the coworking space industry. These different interpretations of sustainability suggests that there are multiple types of sustainable coworking space business models. By identifying these four perspectives, we move a step closer to being able to assess the sustainability of coworking space business models. This is important as sustainability is often claimed in the industry, yet a lack of transparency exists as to what the meaning of sustainability is.
This research offers four approaches to a sustainability coworking space.”
- Line 590 to 593:
“Firstly, it would be valuable to further research the four perspectives identified, to place them into context, and to move towards assessing them sustainability wise. As these perspectives are of subjective nature, it will be important to move towards an objective sustainability assessment.”
- Line 596 to 597:
“Lastly, there is a clear need for an integrated sustainability framework in the coworking space context.”
Reviewer 2 Report
This article offers some interesting insights into how a sample involved in co-working values different aspects of sustainability. The study has value to those interested in advancing sustainability in co-working spaces, recruiting people to different co-working opportunities, and better understanding how co-working might advance certain sustainability goals and values. I think you do a nice job of interpreting and explaining the data. One thing I would note is you indicated that the sample you chose was one that included people who were recruited in part because they had an interested in sustainability to begin with - as a result, you might want to qualify the findings as representing a subset of co-workers, as a sample that had not sought out sustainability-minded folks might have different results. There were a few places where the grammar or word choice was awkward but overall this was well written and clearly presented. I had a few comments in the attached, and highlighted some places where the grammar might need attention.

Author Response
Dear Reviewer,
Thank you for taking the time to help us improve our paper. We are pleased you enjoyed reading it and are very grateful for your help. We submitted one manuscript with highlighted changes and one without. Below is our point-by-point response. Thank you again!
“The study has value to those interested in advancing sustainability in co-working spaces, recruiting people to different co-working opportunities, and better understanding how co-working might advance certain sustainability goals and values. I think you do a nice job of interpreting and explaining the data.”
Thank you, we truly appreciate your comments.
“One thing I would note is you indicated that the sample you chose was one that included people who were recruited in part because they had an interested in sustainability to begin with - as a result, you might want to qualify the findings as representing a subset of co-workers, as a sample that had not sought out sustainability-minded folks might have different results.”
This is an excellent point. Thank you for pointing this out. We changed the following sections accordingly:
- Line 218 to 221
“This was done by reaching out to coworking spaces with specific sustainability mission statements. As a result, this sample can be considered as one that is skewed towards the belief of the importance of sustainability within the coworking context.” - Line 505 to 508
“This study offers an insight into how coworking spaces view sustainability, particularly coworking spaces that put an emphasis on the topic of sustainability, as participants of this study felt strongly about the importance of sustainability in the coworking space context.” - Line 585 to 588
“Lastly, it is important to consider that the sample selected had a bias towards the importance of sustainability. A different sample may have yielded different results, yet the choice of sample was chosen in this manner to extract different subjective perspectives on what a sustainable coworking space is.”
“There were a few places where the grammar or word choice was awkward but overall this was well written and clearly presented.”
Thank you! We appreciate your help fixing these minor mistakes.
“I had a few comments in the attached, and highlighted some places where the grammar might need attention.”
Thank you for taking the time to go through these and help us. Here are all the corrections listed (apart from simple grammar and wording):
1. Changed “concern” to: “once again illustrating that there may be multiple types of sustainable business models within the context of coworking spaces.”
- Changed “ICT-mediated” to: “Firstly, the sharing economy is mediated by Information and Communications Technology. This means that technology offers the opportunity for the sharing economy to be scalable”
- Changed “yet there is no homogenous approach to sustainability observable” to: “yet there does not seem to be a homogenous approach”.
- Changed “to develop a greater understanding of sustainability aspects and practices within the industry.” to “Thus, exploratory research is needed to better understand the sustainability aspects of coworking spaces and to move towards an objective sustainability assessment of them.” (Changed the section in addition to this due to the comments of another reviewer)
- “Second, 42 written narratives from coworking spaces about sustainability were examined.” Added this section to better explain:
“These were retrieved from the websites of coworking spaces, as well as social media and blog posts made by coworking spaces.” - Explained “to help participants in the overall sort.”, by adding:
“this stage was included to help participants get an overview over all Q-Sort statements before proceeding with the main sort. This is a common step in Q-methodology to maximize the quality of the Q-Sort.”
Round 2
Reviewer 1 Report
Improved a lot. Can be published.
Author Response
Thank you very much for all your help. We truly appreciate it!
This manuscript is a resubmission of an earlier submission. The following is a list of the peer review reports and author responses from that submission.